# Mechanisms of TDP-43 Proteinopathy Onset and Propagation

**DOI:** 10.3390/ijms22116004

**Published:** 2021-06-02

**Authors:** Han-Jou Chen, Jacqueline C. Mitchell

**Affiliations:** 1Department of Biology, York Biomedical Research Institute, University of York, Wentworth Way, York YO10 5DD, UK; 2Maurice Wohl Clinical Neuroscience Institute, Institute of Psychiatry, Psychology and Neuroscience, King’s College London, 125 Coldharbour Lane, Camberwell, London SE5 9NU, UK; jacqueline.mitchell@kcl.ac.uk

**Keywords:** TDP-43, neurodegeneration, ALS, frontotemporal dementia, proteinopathy

## Abstract

TDP-43 is an RNA-binding protein that has been robustly linked to the pathogenesis of a number of neurodegenerative disorders, including amyotrophic lateral sclerosis and frontotemporal dementia. While mutations in the TARDBP gene that codes for the protein have been identified as causing disease in a small subset of patients, TDP-43 proteinopathy is present in the majority of cases regardless of mutation status. This raises key questions regarding the mechanisms by which TDP-43 proteinopathy arises and spreads throughout the central nervous system. Numerous studies have explored the role of a variety of cellular functions on the disease process, and nucleocytoplasmic transport, protein homeostasis, RNA interactions and cellular stress have all risen to the forefront as possible contributors to the initiation of TDP-43 pathogenesis. There is also a small but growing body of evidence suggesting that aggregation-prone TDP-43 can recruit physiological TDP-43, and be transmitted intercellularly, providing a mechanism whereby small-scale proteinopathy spreads from cell to cell, reflecting the spread of clinical symptoms observed in patients. This review will discuss the potential role of the aforementioned cellular functions in TDP-43 pathogenesis, and explore how aberrant pathology may spread, and result in a feed-forward cascade effect, leading to robust TDP-43 proteinopathy and disease.

## 1. Introduction

RNA-binding proteins (RBPs) are heavily implicated in neurodegenerative diseases such as amyotrophic lateral sclerosis (ALS), frontotemporal dementia (FTD) and Alzheimer’s disease (AD). Mutations in RBPs such as TDP-43, FUS, MATR3, TIA1 and ATXN2 are linked to inherited forms of ALS, and aberrant aggregation of TDP-43 or FUS is observed in affected neurons in the majority of ALS and FTD cases [1]. Amongst the RBPs linked to neurodegeneration, TDP-43 has received the most attention, as pathogenic accumulations of the protein are found in the majority of ALS patients, as well as tau-negative FTD and a proportion of AD [2,3], and have been shown to promote neurodegeneration in cell and animal models [4,5]. There are a number of reviews already in existence focusing on exactly how TDP-43 pathology may lead to neuronal cell death, hence in this review, we will focus on current knowledge regarding cellular processes that may be triggers for TDP-43 pathology initiation and how this pathology propagates throughout the CNS leading to global neurodegeneration. 

## 2. TDP-43 in Health

Identified in 1995 as a repressor which binds to the trans-active response element DNA sequence of the HIV-1 viral genome, TDP-43 is a ubiquitously expressed DNA/RNA-binding protein that is predominantly localised to the nucleus but can shuttle between the nucleus and cytoplasm [6,7]. It binds and regulates UG-rich RNAs and TG-rich single stranded DNA, impacting on functions such as splicing, transportation and degradation [8,9,10], and thereby plays a key role in local protein translation in neuronal axons and neurites [11,12]. Like many RBPs, TDP-43 can also form cytoplasmic stress granules (SGs) in response to cellular stress, which safeguards the essential mRNA from degradation and promotes rapid recovery when the stress is removed [13]. Crucially, homozygous TDP-43 knockout mice are not viable indicating it is an essential protein for development and survival [14].

## 3. The Triggers of TDP-43 Proteinopathy

Disease-associated mutations in TDP-43 account for roughly 3% of familial ALS [15]. However, TDP-43 is identified as one of the major components of intracellular protein aggregates in neuronal tissues in the majority of ALS cases including cases caused by mutations in most ALS-linked genes (except for SOD1 or FUS), and sporadic ALS which has no apparent genetic cause [2,16,17]. This highly phosphorylated, ubiquitinated, and detergent-insoluble cytoplasmic aggregation of TDP-43 linked to neurodegeneration is known as TDP-43 proteinopathy. A growing body of evidence now suggests that disturbances of various cellular functions and exposure to cellular stress may contribute to the build-up of TDP-43 proteinopathy and hence may be instrumental factors in the onset and progression of ALS pathogenesis. 

### 3.1. Nucleocytoplasmic Transport

TDP-43 contains both a nuclear localisation signal (NLS) and nuclear export signal (NES) which are recognised by importin and exportin proteins respectively, allowing TDP-43 to shuttle between the nucleus and cytoplasm. Nuclear import of TDP-43 is mediated by importin-α [18], whereas a number of exportins are suggested to be responsible for TDP-43 nuclear export [19,20,21]. As TDP-43 pathology is formed of cytoplasmic aggregation accompanied by clearance from the nucleus, it was suggested early on that a re-distribution of TDP-43 could be the cause of its aggregation. Indeed, disease-linked mutations at the C-terminal prion-like domain (PLD) are found to enhance TDP-43 cytoplasmic localisation [22,23,24]. Cytoplasmic TDP-43 expression subsequently leads to a disrupted nuclear envelope and impaired nucleocytoplasmic transport, components of which are enriched in the interaction portfolio of cytoplasmic insoluble TDP-43 aggregates [25]. Reduced expression of functional TDP-43 also leads to reduced Ran GTPase, which could potentially compromise nucleocytoplasmic transport via its role in regulating the nucleocytoplasmic shuttling mediated by importin and exportin proteins [26,27]. Inhibition of nucleocytoplasmic transport subsequently triggers the formation of TDP-43 cytoplasmic aggregation accompanied by increased cell death [28]. The disruption of nucleocytoplasmic transport in ALS is further evidenced by the presence of nuclear pore pathology and reduced protein levels of importin-α subunits observed in the brain tissues of ALS/FTD patients carrying TDP-43 proteinopathy [18,25]. These findings illustrate a positive correlation where disruption of nucleocytoplasmic transport promotes cytoplasmic localization of TDP-43, which in turn interacts and potentially displaces components involved in nucleocytoplasmic transport, further compromising the machinery. 

Elevated cytotoxicity is observed in cells or animal models expressing cytoplasmic TDP-43 [4,29,30], whereas preventing TDP-43 nuclear export either via mutating the NES, or using pharmacological treatment to inhibit exportins, partially reduces TDP-43 cytotoxicity [20,30], indicating that cytoplasmic TDP-43 could promote TDP-43 cytotoxicity. However, both importin knockdown and mutation of the NLS cause cytoplasmic TDP-43 distribution without inducing the formation of aggregates [18] suggesting that cytoplasmic mislocalisation per se is not sufficient to cause TDP-43 proteinopathy. 

### 3.2. Protein Homeostasis

TDP-43 is an aggregation-prone protein under strict autoregulation where TDP-43 protein binds to the 3’UTR of its transcript to prevent it being translated, thereby maintaining a stable intracellular protein concentration [31,32]. Disease-causing mutations in TARDBP have been shown to enhance TDP-43 protein half-lives and aggregation propensity [33,34,35]. Furthermore, elevated TDP-43 protein levels, either through protein overexpression or through inhibition of protein degradation machinery leads to TDP-43 proteinopathy and cytotoxicity in both cell and animal models [5,36,37]. Increased total TDP-43 protein levels are also observed in motor cortex and spinal cord tissues as well as the plasma and cerebrospinal fluid (CSF) of ALS patients, which are positively correlated with disease severity and duration [38,39,40,41], suggesting the disruption of TDP-43 homeostasis contributes to TDP-43 proteinopathy and disease development. 

While TDP-43 protein production is regulated via an autoregulatory mechanism, other cellular machineries are involved in maintaining TDP-43 protein homeostasis. In particular, chaperone proteins including HSP70, DNAJB8 and DNAJB2a are known to interact with intracellular TDP-43 to facilitate its folding, refolding and the formation of reversible stress-induced granules [5,42,43] (Figure 1). Consistent with this, inhibition of chaperone activity enhances TDP-43 aggregation and cytotoxicity in cells [5].

The final safeguard of protein homeostasis regulation is via protein degradation. It is known that TDP-43 can be degraded via three pathways: the ubiquitin proteasome system (UPS), the autophagy lysosome pathway, and chaperone mediated autophagy (CMA). The UPS has been shown to mediate the degradation of soluble full length TDP-43 as well as C-terminal fragments [44,45,46,47] whereas autophagy plays a more important role in the degradation of TDP-43 aggregates potentially through a VCP-mediated endocytic pathway [44,45,46,47,48,49]. CMA is a form of autophagy which is facilitated by chaperone protein Hsc70 to bring the substrate to the lysosome. As Hsc70 is shown to interact with aggregated TDP-43 protein and endogenous TDP-43 is observed to localise to CMA-lysosomes, this indicates that TDP-43 is a substrate for CMA-mediated degradation [45,50]. Whereas inhibition of UPS or autophagy triggers TDP-43 accumulation and aggregation, enhancing the degradation machineries facilitates TDP-43 protein turnover and reduces TDP-43 cytotoxicity [51,52,53,54,55] (Figure 1).

To further support the importance of altered function of the protein degradation machineries in TDP-43 proteinopathy development, a number of TDP-43 proteinopathy causative mutations have been identified in genes linked to autophagy or the UPS, including p62, optineurin (another autophagy receptor) [56,57], TANK-binding kinase 1 (TBK1) [58,59] which phosphorylates both p62 and optineurin [60], and valosin-containing protein (VCP) [61] and ubiquilin 2 [62], both of which are required for normal UPS and autophagy function [63,64,65]. Of particular note, TDP-43 pathology in patients with these mutations is indistinguishable from that reported in sporadic cases, or in those with TARDBP mutations.

Compared to other cell types, neurons are particularly susceptible to the build-up of aberrant protein aggregates due to their highly specified structures, often with axon and dendritic terminals some distance from the main cell body, particularly in the case of motor neurons. This increased susceptibility means these cells are likely to place an increased demand on chaperone proteins and protein degradation machineries. However, motor neurons have a higher threshold for chaperone activation [66], which, accompanied by a gradual reduction of HSF1, the master transcription factor controlling chaperone protein expression [67,68], and proteasome activity [69,70] with age, exposes aged motor neurons to a greater risk of mis-folded protein accumulation caused by insufficient chaperoning. Indeed, reduced chaperone levels are found in the spinal cord of ALS patients and TDP-43 transgenic mice [5] suggesting that disruption of the chaperone system as well as reduced protein degradation efficiency may be early events which subsequently lead to the build-up of TDP-43 proteinopathy and disease development.

### 3.3. RNA Interaction

Although RBPs have been implicated in multiple neurodegenerative diseases, and the RNA splicing, regulation and trafficking mediated by these RBPs are demonstrated to play important roles in neuronal health, direct evidence linking TDP-43 RNA-binding to proteinopathy and disease development is scarce and ambiguous. 

It has been shown that in addition to protein chaperones, nucleic acids can also act as chaperones to prevent nucleic acid-binding proteins from aggregating and to enhance protein solubility [71]. As a DNA/RNA binding protein, the binding of TDP-43 to single stranded DNA or RNA effectively prevents TDP-43 aggregation [72,73,74]. Nevertheless, as there is no evidence indicating that disease-causative mutations in the PLD impact on TDP-43’s capacity to bind to RNA [74] and RNAs are not found in TDP-43 aggregates in ALS/FTD post-mortem tissue [75], it is currently unclear whether RNA-binding of TDP-43 is disrupted and may subsequently contribute to proteinopathy and disease development. However, a growing number of recent studies have highlighted the potential importance of RNA-binding in TDP-43 aggregation. 

Currently, there are only three ALS/FTD-linked mutations identified in or around TDP-43 RNA binding motifs (RRMs) that abolish the RNA interaction [74,76]. Patients carrying the RNA-binding mutant TDP-43 display classic TDP-43 proteinopathy [74,77,78]. Disruption of RNA-binding either by the disease-causative RNA-binding deficient mutations or by engineered nucleic acid substitutions disrupting TDP-43 RNA-interactions, leads to TDP-43 protein being more prone to forming nuclear aggregates [7,74,75,79,80]. Current understanding indicates that these nuclear TDP-43 aggregates caused by the loss of RNA-interactions are distinctly different from SGs and may not be exactly the same as pathological TDP-43 aggregates. However, they can be transformed to more fixed gel-like aggregates when exported to the cytoplasm or when certain chaperone activity is reduced [80]. In additional to the direct impact on TDP-43 protein aggregation, RNA-interactions also play an important role in mediating other TDP-43 cellular functions such as its recruitment into SGs and interaction with other proteins [75,80,81,82]. Interestingly, cytotoxicity is only observed in disease-causing RRM mutants not the engineered RNA-binding deficient mutant TDP-43 [79,83], whether this indicates an effect on protein expression levels, or whether disease-causing RRM mutations cause other disruptions which trigger TDP-43 toxicity either jointly with or independently of the loss of RNA-binding still waits to be clarified. One important finding supporting a role of RNA interaction in TDP-43 proteinopathy is a cell study which found that the addition of RNA not only reduces TDP-43 aggregation, but also rescues TDP-43 cytotoxicity in cultured neurons over expressing wild type TDP-43 [75], demonstrating the potential of RNA as a chaperone for RBPs like TDP-43. However, further research is warranted to validate this function and to reveal the role of RNA-binding during the build-up of TDP-43 proteinopathy.

### 3.4. Cellular Stress

Following the link between RBPs and neurodegenerative disease, an array of cellular studies have demonstrated that many RBPs including TDP-43 can be recruited to SGs in response to a variety of stressors (Table 1). This led to the hypothesis that SGs may form a crucible for ALS pathogenesis, specifically bringing TDP-43 molecules into close proximity, increasing the risk of initiating aggregate formation [84]. This hypothesis has been strengthened by studies demonstrating that core SG markers such as PABP1, TIA1 and eIF3 have been identified within TDP-43 aggregates in ALS and FTD patient tissue [85,86,87], despite an element of disagreement from more recent studies [88]. 

Stress granules form when translation initiation is impaired, typically in response to a variety of stressors, such as heat, ER stress, proteasome inhibition, oxidative stressors such as arsenite, and osmotic stressors such as sorbitol (for review see [89]). A number of proteins and RNAs are recruited to these granules, which contain translation initiation complexes, and it has been hypothesized that RNA bound TDP-43 is recruited to these granules at least in part to preserve its RNA cargos ready for translation at the end of the stress exposure. With a few rare exceptions, the majority of studies have demonstrated that TDP-43 is recruited to SGs during cellular stress (Table 1), however, clear links between this recruitment and the formation of pathogenic TDP-43 aggregates are yet to be proven. 

**Table 1 ijms-22-06004-t001:** A summary of cell studies that explicitly demonstrate the recruitment (or lack) of TDP-43 to stress granules identified by one or more of the key SG markers, G3BP, TIA1, TIAR, HuR, eIF3 and FMRP. Only papers that explicitly assessed TDP-43 recruitment to granules are included, even if this was not the main thrust of the paper. Any publications which failed to show clear SG formation using a core SG marker, irrespective of their TDP-43 findings, are not included. With rare exceptions, the majority of findings support the hypothesis that irrespective of stress or cell type, if robust SGs form, TDP-43 is recruited to them. Endog = endogenous; Exog = exogenous, WT = wild type. † TDP-43 antibody-dependent.

Cell Type	Stressor	SG Marker	TDP Recruitment to SGs	Type of TDP	Ref.
Fibroblasts	Arsenite	TIAR	chronic (30 h) only	Endog WT and A382T	[90]
		HuR/TIA1	N	Endog WT and A382T	[91]
Fibroblasts	H_2_O_2_	TIAR	Y	Endog WT, A382T, G294V	[92]
COS-7	4-Hydroxynonenal	TIAR	Rare but persistent	Endog WT	[93]
COS-7	Arsenite	G3BP1/TIAR/	Y	Exog WT-YFP	[94]
		eIF3			
COS-7	Heat Shock	G3BP1	Y	Exog WT-GFP	[94]
HeLa	Paraquat	HuR	Y	Endog WT	[95]
		HuR	Y	Endog WT	[96]
		HuR/TIAR	Y	Endog WT	[97]
		HuR	Y	Endog WT	[98]
HeLa	Arsenite	HuR	Y	Endog WT	[95]
		HuR	Y	Endog WT	[96]
		HuR	Very Rare	Endog WT	[99]
		HuR	Y	Endog WT	[98]
		TIA1	Y	Endog WT	[100]
		TIAR/G3BP	Y	Endog WT/Exog HA-WT	[30]
		TIA1/G3BP1	Y	Endog WT	[101]
		TIA1	Y	Endog WT	[102]
		TIA1	Y	Exog V5-TDP WT/NLS	[86]
		G3BP1	Y	Exog WT, NLS, G348C	[103]
HeLa	Heat shock	TIA1	Y	Endog WT	[102]
		G3BP1	Y	Exog WT, NLS, G348C	[103]
		TIA1	Y	Exog V5-TDP WT/NLS	[86]
HeLa	Thapsigargin	TIA1	Y	Endog WT	[102]
		HuR	Y	Endog WT	[99]
HeLa	Clotrimazole	TIA1	Y	Exog V5-TDP WT/NLS	[86]
Hek293T	Sorbitol	TIAR/HuR	Y	Endog WT	[104]
		eIF3/TIAR	Y	Endog WT	[105]
		TIAR	Y	Endog WT	[106]
Hek293T	Arsenite	TIAR	N	Endog WT	[104]
		FMRP	Y	Exog NLS	[88]
		eIF3/TIAR	Y	Endog WT	[105]
Hek293T	H_2_O_2_	TIAR	Y	Exog WT	[107]
U2OS	Arsenite	eIF3η	Y	Endog WT	[108]
		TIAR	Y	Endog WT	[109]
U2OS	Optogenetic	Opto-G3BP/A11	Y	Endog WT (also pTDP)	[109]
U2OS	Arsenite	G3BP/TIAR	Y^†^	Endog WT	[110]
SH-SY5Y	Arsenite	G3BP	Y	Exog WT/G348C/NLS	[103]
SH-SY5Y	Heat shock	G3BP	Y	Exog WT/G348C/NLS	[103]
SH-SY5Y	Paraquat	HuR	Y	Endog WT	[95]
		HuR	Y	Endog WT	[97]
		HuR	Y	Endog WT	[96]
		HuR/TIAR	Y	Endog WT	[98]
		HuR	Y	Endog WT	[111]
U87MG	Paraquat	HuR	Y	Endog WT	[95]
astroglia					
BE(2)-M17	Arsenite	TIA1	Y	Exog eGFP-WT/Q331K/Q343R	[85]
neuroblastoma					
NSC34	Arsenite	TIAR/HuR	Y	Endog WT	[81]
		HuR	Y	Endog WT	[91]
NSC34	Heat shock	TIAR/HuR	Y	Endog WT	[81]
H4 neuro-	Arsenite	G3BP1	Y	Exog GFP-NLS	[82]
glioma					
H4 neuro-	Thapsigargin	G3BP1	Y	Exog GFP-NLS	[82]
glioma					
Primary Glia	Sorbitol	TIAR/HuR	Y	Endog WT	[104]
Primary	Arsenite	TIAR	Y	Exog GFP-WT	[94]
neurons					
Primary	Heat shock	TIA1	Y	Exog V5-TDP WT/NLS	[86]
neurons					
iPSC	Optogenetic induction	Opto-G3BP	Y	Endog WT (also pTDP)	[109]
neurons					
iPSC	Arsenite	TIA1	Y	Endog WT	[109]
neurons		TIAR	Y (chronic only)	Endog WT and A382T	[90]
iPSC	Heat shock	TIA1	Y	Endog WT	[109]
neurons					
iPSC motor	Puromycin	G3BP1	Y	Endog WT/N325S	[82]
neurons					
Mouse brain	Arsenite	TIAR	Y	Endog WT	[112]
slices					

Cell and stress type variability may be of key importance in unpicking the role of the cellular stress response and SGs in the formation of pathogenic TDP-43 aggregates as profound differences in SG dynamics are observed, dependent on the cell and stress type used. Specifically, cortical neurons appear to be somewhat resistant to SG formation compared to other non-neuronal cell types. However, once granules form, they persist for much longer [113]. If SGs do indeed serve as crucibles for pathogenesis, this cell specific increase in their longevity could underpin the selective vulnerability of neurons to TDP-43 pathology. In support of this, ALS/FTD-linked mutations impacting the low complexity domain of the SG protein TIA1 have been reported to alter SG dynamics, specifically prolonging recovery times [101], increasing the likelihood of the transformation of the SGs into fixed gel-like inclusions.

Recent studies however have thrown doubt on the role of SGs in TDP-43 pathogenesis. Stress-induced liquid-liquid phase separation (LLPS) of TDP-43 has been observed in both the nucleus and cytoplasm independently of classic SG makers [28]. As discussed above, the interaction of TDP-43 with RNA reduces TDP-43 aggregation and cytotoxicity, thus SG-associated TDP-43 may be less prone to aggregation than its non-RNA bound counterpart, since the recruitment of TDP-43 to SGs is primarily considered to be an RNA dependent event. Indeed, acute recruitment to SGs protects TDP-43 from aberrant phosphorylation [94], which is believed to be a key feature of the aggregated protein [114]. However, longer term stress apparently causes SGs to resolve, but leaves behind TDP-43 aggregates which are further phosphorylated [94]. Indeed, longer term stress has been reported to result in the exclusion of TDP-43 from stress granules [100]. What happens to this SG associated TDP-43 is currently unclear; whether it is expelled from SGs to be degraded, or otherwise processed, or whether these longer term SGs explicitly do not recruit TDP-43 is yet to be addressed. What is clear however, is that this longer-term stress results in increases in insoluble, phosphorylated TDP-43 [100], implicating the cellular stress response in the formation of pathogenic protein.

To date, the bulk of this work has been conducted in isolated cellular systems, with evidence from in vivo studies being very limited. However, traumatic brain injury in drosophila has been shown to induce the formation of TDP-43 positive SGs, although TDP-43 recruitment to these granules appear to dissipate after 24 hours, suggesting a transient association [115]. In contrast, traumatic brain injury to mice triggered the formation of multiple pTDP-43 granules, which disappear over the course of several days, These granules were rarely co-labelled with the SG marker TIAR although their formation is enhanced in TDP-43 transgenic animals [116], suggesting they may not be part of a classic SG structure, but could represent a granule population at high risk of forming pathogenic aggregates. 

Clearly there is a wealth of evidence demonstrating a possible role for cellular stress in TDP-43 pathogenesis. What is less clear at present, is whether this is linked to the formation of classic SGs, and potentially their persistence beyond the short term, or whether it is actually associated with the stress driven phase separation of non-RNA bound TDP-43, that explicitly does not get recruited to SGs. A recent study has demonstrated that impairment of SG assembly impairs the formation of TDP-43 foci in both acute and chronic stress paradigms, but crucially, did not completely prevent their formation, particularly in the chronic model, suggesting that while SGs may facilitate TDP-43 aggregation, TDP-43 proteinopathy can still develop through a SG-independent mechanism [105,110].

There is also still a dearth of evidence detailing the cellular stress response in vivo. The time course of granule formation and dissolution reported in mice in response to brain injury is markedly longer than that reported in drosophila, or indeed in cell models, suggesting that there may be distinct differences between the cellular stress response in mammals in vivo and that reported in cells, hence further research is still required.

## 4. The Propagation of TDP-43 Proteinopathy

Once TDP-43 pathogenesis has been initiated, there are still a number of questions relating to how this pathology propagates and results in cellular dysfunction and death. Symptom onset in ALS patients tends to be asymmetrical and localised to a specific region of the body. As the disease progresses, symptoms appear to spread outwards from this initial site to ultimately affect the whole body [117]. A similar pattern of TDP-43 pathology progression has also been reported in FTD patients [118], providing strong evidence for a cell-to-cell mechanism underlying the propagation of pathogenic protein. 

### 4.1. TDP-43 Cell-to-Cell Propagation

In vitro studies have demonstrated that both wild-type and disease mutant TDP-43 and its C-terminal fragments can oligomerise [112,119,120] and form amyloid like fibrils [121,122], which are present in patient brains [123]. These aggregated or oligomeric forms of TDP-43 can recruit soluble native TDP-43 [36,124] and seed protein aggregation in otherwise healthy cells, including a concomitant loss of nuclear TDP-43 [28,125,126,127,128,129,130,131], supporting the notion that TDP-43 pathogenesis can spread throughout the CNS from a localised site of initiation. 

Cellular studies have suggested that TDP-43, including oligomeric forms, can be transmitted between cells, providing a route for pathology spread [106,132,133]. This finding is supported by in vivo studies showing that the injection of patient-derived brain extracts can seed TDP-43 pathology in mouse brain, which proceeded to spread to more distal parts of the connected CNS [134], however, the mechanism by which this occurs is not fully established. TDP-43 has been reported to be present in purified microvesicles/exosomes, [106,130], which makes it more efficiently taken up by recipient cells than non-exosome associated TDP-43, suggesting pathology transmission may be driven by exosome secretion. This is supported by findings in vivo showing that TDP levels in exosomes harvested from ALS patient brains are increased, and treatment of N2a cells with these exosomes can induce cytoplasmic redistribution of TDP-43 [131]. Studies in neurons have shown that the protein can be transported both anterogradely and retrogradely along the axon, and across synaptic terminals [106], providing a pathway for the systematic spread of exosomally contained TDP-43. However, in other studies, no uptake of TDP-43 from extracellular vesicles was observed, suggesting direct physical contact between cells is required [133], thus further study is required to clarify the route of transmission. It is possible that both direct contact and exosome secretion may have a role to play in TDP-43 pathology spread, and that some form of ‘priming’ of the recipient cell may impact on how readily this spread occurs. Certainly in vivo studies have suggested that prior mislocalisation of TDP-43 to the cytoplasm is a key factor in the efficient seeding and spread of the pathogenic protein [134]. Since it has been reported that ALS is a multistep process [135], perhaps the combination of a ‘seed’ coupled with a secondary ‘hit’ that makes the recipient cell more vulnerable to TDP-43 pathogenesis could be a trigger for the propagation of pathology.

It is important to note however, that not all cell studies support the notion of TDP-43 propagation. Two reports have shown that while cells may release TDP-43 into the media, there is no evidence of uptake of this protein into recipient cells treated with the conditioned media [136,137]. The conditioned media did result in cytotoxicity and metabolic changes [137], suggesting that TDP-43 may have extracellular impacts. Clearly, more work is required to fully understand the mechanism of spread of TDP-43 pathology, as well as identifying potential inter-versus extracellular effects.

### 4.2. Non-Cell Autonomous Effects

Glial cells provide important supportive functions to motor neurons and are increasingly being recognised as playing key roles in neurodegenerative diseases generally. Astrocytes have been shown to contribute to glutamate excitotoxicity, which has long been implicated in ALS, although more recent studies suggest a more general altered inflammatory response [138]. Oligodendrocytes derived from ALS patient tissue all induced motor neuron cell death in co-culture and conditioned media experiments [139]. Microglia are now well known to exist in both neuroprotective and neurotoxic states, and these may play a major contributory role to the disease process across multiple neurodegenerative disorders. (reviewed in [140]). However, to date there are only limited findings so far relating to how these non-neuronal cells may impact on the spread of TDP-43 pathology [125]. 

In vivo studies have demonstrated a protective role for microglia in TDP-43 proteinopathy in an inducible model of disease. Microglia were relatively unaffected by aberrant TDP-43 expression and were able to rapidly clear neuronal accumulations once the aberrant protein expression was halted [141]. Studies in zebrafish have shown that microglial cells phagocytose TDP-43 within neurodegenerating motor neurons [142], and crucially, in the absence of functional microglia, TDP-43 fragments were released into the extracellular environment from the degenerating neuron. It is possible therefore, that microglial dysfunction may contribute to TDP-43 pathology spread via a failure in the clearance of pathogenic aggregates and degenerating neurons.

An assessment of patient tissue has suggested that oligodendrocytes do not play a major role in the transmission of TDP-43 pathology. HyperphosphorylatedTDP-43 inclusions were only found in the white matter in subcortical regions directly associated with the motor and sensory cortices, with an absence of any overt pathology in axonal pathways [143]. However, a recent study has suggested that TDP-43 and p62 aggregates in ALS patient oligodendrocytes are a major pathogenic burden [144], thus more work is required to fully understand the extent to which these cells are involved in pathology progression.

Investigations into the role of non-neuronal cells in the pathogenesis and progression of TDP-43 linked disease are still comparatively in their infancy, but clearly they have the capacity to play a crucial role in the disease course, and further study will elucidate exactly how these cells interact with both TDP-43 and neurons to impact on disease pathogenesis.

## 5. Concluding Remarks

Since its identification 15 years ago, we have made significant progress in understanding the importance of TDP-43 proteinopathy in neurodegeneration, and the factors contributing to TDP-43 aggregation in cells. It is now clear that this is a tightly controlled system where the disruption of a small part could lead to TDP-43 dysregulation, resulting in a feed-forward cascade effect, further impairing the cellular machinery responsible for maintaining TDP-43 function and homeostasis. Accumulation of excess or aberrantly localised TDP-43 leads to protein aggregation which can spread from cell to cell, and also results in further mislocalisation of TDP-43 from the nucleus. This process is likely to involve not only neuronal cells, but at least in part be influenced by a concomitant dysfunction in supporting glial cells, particularly astrocytes and microglia. As TDP-43 proteinopathy is a major hallmark for age-dependent neurodegenerative diseases, it is not surprising that the activity of many important players involved in maintaining normal TDP-43 function are found to be reduced in aged neurons. Since multiple pathways are functionally interlinked and appear to contribute to TDP-43 proteinopathy, they may provide various potential therapeutic intervention angles to rectify TDP-43 aggregation and thereby halt or even rescue disease progression.

## Figures and Tables

**Figure 1 ijms-22-06004-f001:**
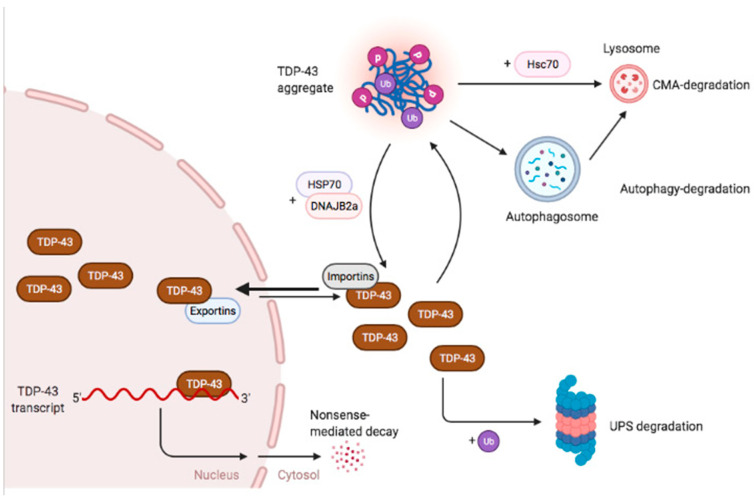
Regulatory mechanisms involved in maintaining cellular TDP-43. The production of TDP-43 is under strict autoregulation where nuclear TDP-43 protein binds to the 3′UTR of its transcript and triggers alternative splicing which leads to nonsense-mediated decay. Cytoplasmic TDP-43 may form ubiquitinated phosphorylated aggregates when the protein concentration is elevated or when the cells are stressed. These aggregates can be resolved by chaperone proteins HSP70 and DNAJB2, or can be degraded via autophagy or CMA. Soluble cytoplasmic TDP-43 is largely degraded via the UPS. The figure was created with BioRender.com.

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
