# Peer review of "Mechanisms of TDP-43 Proteinopathy Onset and Propagation"

_ijms, 2021, doi:10.3390/ijms22116004_

Round 1

Reviewer 1 Report

This review by Chen and Mitchell presents several key aspects of TDP-43 proteinopathy that require heavy study, those being nucleocytoplasmic transport, protein homeostasis, RNA binding, cellular stress, aggregate propagation, and TDP-43-mediated non-cell autonomous pathology. These cellular processes seem to be heavily involved in ALS pathogenesis, and TDP-43 appears to be involved in all of them. Therefore, TDP-43 is a critical protein to study in order to make progress in comprehending the molecular events of ALS. Overall, I found the manuscript to be already quite well organized and illustrated. I did, however, find areas in which further development seemed necessary to improve the impact of the manuscript. Below, I have detailed my suggestions.

Abstract:

  1. line 16: change "fore" to "forefront"
  2. lines 11-13: I found this statement to be ambiguous. The TARDBP gene is mutated at similar frequencies in familial and sporadic cases, 4% and 1% respectively (Daoud et al., 2009; Millecamps et al., 2011; Muller et al., 2018). So, what do the authors mean about the pathology arising sporadically? Do they mean that wtTDP-43 might be involved in driving pathogenesis in sporadic ALS cases? It just felt that the authors were comparing apples to oranges in this sentence, and the abstract should be as clear as possible.

3.1 Nucleocytoplasmic transport

  1. As the authors mention that nuclear import of TDP-43 is mediated by importin-alpha, it would also be interesting to the reader, in my opinion, to add that Ran GTPase als regulates import and export of cargo. Of note, TDP-43 is also responsible for regulating the mRNA of Ran GTPase. This is another reason for which nucleocytoplasmic transport is hindered by TDP-43 proteinopathy (Ward et al., 2014; J Exp Med).

3.2 Protein homeostasis

  1. In the paragraph explaining that a number of causative mutations have been identified in genes linked to autophagy or the UPS, it would be effective to elaborate some more on this. For example, all of the genes mentioned have been found to be mutated in a certain frequency of ALS cases, both familial and sporadic. Following that these mutations can cause TDP-43 proteinopathy, could the authors specify which components of the proteinopathy are reproduced? All of them, meaning that mutating TBK1, for example, can lead to TDP-43 cytoplasmic accumulation, aggregation, and toxicity?

3.3 RNA interaction

  1. Indeed, the cellular consequences of the re-solubilization of TDP-43 through RNA are heterogeneous; some studies show a protective effect, while others show a toxic effect. Therefore, the authors could add that further studies on this subject should study the effects of TDP-43 binding with multiple types of RNA in terms of their sequence (many UG repeats?), 3D structure (G quadruplex?), length, etc.

4.1 TDP-43 cell-to-cell propagation

  1. This is a very interesting topic, because we still do not know how relevant the propagation found in cell culture is to the actual disease in patients. I agree with the authors that most likely TDP-43 propagation depends on a two-fold trigger: the seeding species and the secondary hit in recipient cells. Pokrishevsky and others (2016; Sci Rep) found that simply overexpressing TDP-43 in cells lead to the development of pathological SOD-1, which was propagated through conditioned medium. However, this study could not find TDP-43 propagation. This suggests that not only do propagation mechanisms of different proteins seem to differ but also that TDP-43 propagation seems more complicated to orchestrate. Furthermore, Hergesheimer et al (2020; Cells) did not find TDP-43 propagation through conditioned medium either. However, TDP-43 was found in the conditioned medium, which was associated with a particular change in the metabolism of recipient cells. Consequenctly, this study shed light on the idea that TDP-43 could have pathological roles both outside and inside of cells. Therefore, it would be beneficial to distinguish the effects between TDP-43 that is in the extracellular environment and TDP-43 that has successfully entered cells.

Non-Cell Autonomous Effects

  1. It would be beneficial for the article to present some more concrete examples of the involvement of glia in motor neuron degeneration. For example, low levels of the monocarboxylic acid transporter 1 were found in the oligodendrocytes of SOD-1 mice (Lee et al., 2012; Nature). Oligodendrocytes are needed to exchange glycolytic nutrients with motor neurons, and this MCT1 transporter is necessary for that function. Another example is the glutamate excitotoxicity aggravated by astrocytes. If the EAAT2 transporter on the astrocyte membrane is removed or under-expressed, then glutamate cannot be properly eliminated from the synaptic space, causing toxicity to motor neurons that will suffer from increased Ca2+ uptake (Karki et al., 2014; Neurochem Res.). Finally, it is known that microglia exist in an M1 and an M2 form. The former is neurotoxic, while the latter is neuroprotective. Therefore, the balance between these forms is crucial for motor neuron survival. In cell culture, the M1 type has been seen to decrease motor neuron survival, while the M2 type has augmented survival (Liao et al., 2012; Exp Neurol.). Over the course of development of motor decline in mice, there is a parallel increase in the abundancy of the M1 type. It will be paramount to investigate the role of TDP-43 in these pathways.

Author Response

Abstract:

  1. line 16: change "fore" to "forefront"

Thank you, this has been rectified.

  1. lines 11-13: I found this statement to be ambiguous. The TARDBP gene is mutated at similar frequencies in familial and sporadic cases, 4% and 1% respectively (Daoud et al., 2009; Millecamps et al., 2011; Muller et al., 2018). So, what do the authors mean about the pathology arising sporadically? Do they mean that wtTDP-43 might be involved in driving pathogenesis in sporadic ALS cases? It just felt that the authors were comparing apples to oranges in this sentence, and the abstract should be as clear as possible.

Sorry for the lack of clarity, we are referring to the fact that TDP-43 pathology occurs in almost all ALS independently of mutations in the TARDBP gene. It has been observed in ALS patients carrying mutations in TDP-43, VCP, C9orf72, VAPB etc. and also in sporadic ALS carrying no known ALS-linked mutations. We have re-phrased part of the abstract and also the main text (section 3) to avoid confusion as shown:

“TDP-43 proteinopathy is present in the majority of cases regardless of mutation status”

And

“TDP-43 is identified as one of the major components of intracellular protein aggregates in neuronal tissues in the majority of ALS cases including cases caused by mutations in most ALS-linked genes except for SOD1 or FUS, and sporadic ALS which has no apparent genetic cause”

3.1 Nucleocytoplasmic transport

  1. As the authors mention that nuclear import of TDP-43 is mediated by importin-alpha, it would also be interesting to the reader, in my opinion, to add that Ran GTPase als regulates import and export of cargo. Of note, TDP-43 is also responsible for regulating the mRNA of Ran GTPase. This is another reason for which nucleocytoplasmic transport is hindered by TDP-43 proteinopathy (Ward et al., 2014; J Exp Med).

We appreciate the input. Relevant text has now been incorporated into the manuscript, as:

“Reduced expression of functional TDP-43 also leads to reduced Ran GTPase, which could potentially compromise nucleocytoplasmic transport via its role in regulating the nucleocytoplasmic shuttling mediated by importin and exportin proteins (Lui and Huang, 2009; Ward et al., 2014)”

3.2 Protein homeostasis

  1. In the paragraph explaining that a number of causative mutations have been identified in genes linked to autophagy or the UPS, it would be effective to elaborate some more on this. For example, all of the genes mentioned have been found to be mutated in a certain frequency of ALS cases, both familial and sporadic. Following that these mutations can cause TDP-43 proteinopathy, could the authors specify which components of the proteinopathy are reproduced? All of them, meaning that mutating TBK1, for example, can lead to TDP-43 cytoplasmic accumulation, aggregation, and toxicity?

We understand the reviewer’s interest in further details regarding the ALS-linked genes particularly in protein degradation pathway, but this would deviate from the scope of this review which is about TDP-43. In addition, this subject of ALS genetics has already been covered extensively with multiple reviews such as Mathis et al., 2019 and Mejzini et al., 2019

And yes, TDP-43 pathology is indistinguishable between sporadic ALS and familial ALS carrying mutation in ALS-linked genes (except for SOD1 and FUS).

However, we have added the following sentence to clarify that TDP-43 pathology is not altered in patients with any of these mutations:

“Of particular note, TDP-43 pathology in patients with these mutations is indistinguishable from that reported in sporadic cases, or in those with TARDBP mutations.”

3.3 RNA interaction

  1. Indeed, the cellular consequences of the re-solubilization of TDP-43 through RNA are heterogeneous; some studies show a protective effect, while others show a toxic effect. Therefore, the authors could add that further studies on this subject should study the effects of TDP-43 binding with multiple types of RNA in terms of their sequence (many UG repeats?), 3D structure (G quadruplex?), length, etc.

We appreciate the input and have incorporated relevant text in the manuscript:

“However, further research is warranted to validate this function and to reveal the role of RNA-binding during the build up of TDP-43 proteinopathy”

4.1 TDP-43 cell-to-cell propagation

  1. This is a very interesting topic, because we still do not know how relevant the propagation found in cell culture is to the actual disease in patients. I agree with the authors that most likely TDP-43 propagation depends on a two-fold trigger: the seeding species and the secondary hit in recipient cells. Pokrishevsky and others (2016; Sci Rep) found that simply overexpressing TDP-43 in cells lead to the development of pathological SOD-1, which was propagated through conditioned medium. However, this study could not find TDP-43 propagation. This suggests that not only do propagation mechanisms of different proteins seem to differ but also that TDP-43 propagation seems more complicated to orchestrate. Furthermore, Hergesheimer et al (2020; Cells) did not find TDP-43 propagation through conditioned medium either. However, TDP-43 was found in the conditioned medium, which was associated with a particular change in the metabolism of recipient cells. Consequenctly, this study shed light on the idea that TDP-43 could have pathological roles both outside and inside of cells. Therefore, it would be beneficial to distinguish the effects between TDP-43 that is in the extracellular environment and TDP-43 that has successfully entered cells.

We agree that the propagation of TDP-43 is clearly complex, and have added a paragraph to this section to address the lack of propagation in the mentioned studies, and briefly discuss the inter versus extracellular impacts. Since our primary focus is TDP, we have not discussed any specifics regarding SOD-1, as that falls outside the remit of this review. The text we have added is as follows:

“It is important to note however, that not all cell studies support the notion of TDP-43 propagation. Two reports have shown that while cells may release TDP-43 into the media, there is no evidence of uptake of this protein into recipient cells treated with the conditioned media (Porishevsky et al, 2016, Sci Rep, Hegesheimer et al., 2020 Cells). The conditioned media did result in cytotoxicity and metabolic changes (Hegesheimer et al., 2020 Cells), suggesting that TDP-43 may have extracellular impacts. Clearly, more work is required to fully understand the mechanism of spread of TDP-43 pathology, as well as identifying potential inter- versus extracellular effects.”

Non-Cell Autonomous Effects

  1. It would be beneficial for the article to present some more concrete examples of the involvement of glia in motor neuron degeneration. For example, low levels of the monocarboxylic acid transporter 1 were found in the oligodendrocytes of SOD-1 mice (Lee et al., 2012; Nature). Oligodendrocytes are needed to exchange glycolytic nutrients with motor neurons, and this MCT1 transporter is necessary for that function. Another example is the glutamate excitotoxicity aggravated by astrocytes. If the EAAT2 transporter on the astrocyte membrane is removed or under-expressed, then glutamate cannot be properly eliminated from the synaptic space, causing toxicity to motor neurons that will suffer from increased Ca2+ uptake (Karki et al., 2014; Neurochem Res.). Finally, it is known that microglia exist in an M1 and an M2 form. The former is neurotoxic, while the latter is neuroprotective. Therefore, the balance between these forms is crucial for motor neuron survival. In cell culture, the M1 type has been seen to decrease motor neuron survival, while the M2 type has augmented survival (Liao et al., 2012; Exp Neurol.). Over the course of development of motor decline in mice, there is a parallel increase in the abundancy of the M1 type. It will be paramount to investigate the role of TDP-43 in these pathways.

Since the primary focus of this part of our review is the specific role of glial cells in the propagation of TDP-43 pathology rather than a more general contribution of these cells to the disease process, we did not want to expand this section too much, as it would shift our focus. However, we agree that a brief discussion of glial involvement in the disease process would be a useful introduction to this section, and so have added the following text at the beginning of this section:

“Glial cells provide important supportive functions to motor neurons, and are increasingly being recognised as playing key roles in neurodegenerative diseases generally. Astrocytes have been shown to contribute to glutamate excitotoxicity, which has long been implicated in ALS, although more recent studies may suggest a more general altered inflammatory response (Jordan et al., 2018, Front Cell Neurosci.). Oligodendrocytes derived from ALS patient tissue all induced motor neuron cell death in co-culture and conditioned media experiments (Ferraiuolo et al., PNAS 2016). Microglia are now well known to exist in both neuroprotective and neurotoxic states, and these may play a major contributory role to the disease process across multiple neurodegenerative disorders.  (reviewed in Tang and Le, 2015, Mol Neurobiol).

However, to date there are only limited findings so far relating to how these non-neuronal cells may impact on the spread of pathology.....”

We believe this acknowledges the growing importance of glial cells to the disease process, without unduly losing our focus on specifically how they might contribute to the progression of TDP-43 proteinopathy.

Reviewer 2 Report

The review is an excellent one. Well written, very clear, with all the most recent references in the field. The two main topics are very well detailed and discussed.

Minor comment:

Table 1 is a focus on SG granules which is not the main subject of the review, so I don't know if this is relevant to keep it.

Author Response

Minor comment:

Table 1 is a focus on SG granules which is not the main subject of the review, so I don't know if this is relevant to keep it.

We agree that SGs are not the main subject of the review. We included this table because we believe it may be a useful resource for researchers, however, we are happy to remove it if the editor and reviewers think this would benefit the review.

Round 2

Reviewer 1 Report

I thank the authors for taking the time to reply to each of my comments. The current version of the manuscript has, indeed, been sufficiently improved, and I recommend its publication in its current form. Below, I provide two comments regarding wording that the authors might want to modify before publication:

  1. Line 19: Is "healthy" the right term to describe non-pathological TDP-43? The authors might want to change it to "physiological", as "healthy" seems to refer mostly to the state of an organ or organism.
  2. The heading of Fig. 1 appears to be incomplete: "Regulatory mechanisms involved in maintaining"? Should it be, "Regulatory mechanisms involved in maintaining physiological TDP-43 levels"?

Author Response

We would like to thank the reviewer for reading our manuscript and spotting the errors which are now rectified.